# CRP Involved in Nile Tilapia (*Oreochromis niloticus*) against Bacterial Infection

**DOI:** 10.3390/biology11081149

**Published:** 2022-07-30

**Authors:** Qi Li, Baijian Jiang, Zhiqiang Zhang, Yongxiong Huang, Zhou Xu, Xinjin Chen, Jia Cai, Yu Huang, Jichang Jian

**Affiliations:** 1Guangdong Provincial Key Laboratory of Aquatic Animal Disease Control and Healthy Culture, College of Fishery, Guangdong Ocean University, Zhanjiang 524094, China; vickyqi1015@163.com (Q.L.); 15113985614@163.com (B.J.); zzq248798423@gmail.com (Z.Z.); yingxiongh788@gmail.com (Y.H.); xz18728893538@163.com (Z.X.); cxj370265234@163.com (X.C.); matrix924@126.com (J.C.); 2Laboratory for Marine Biology and Biotechnology, Qingdao National Laboratory for Marine Science and Technology, Qingdao 266003, China; 3Guangdong Provincial Engineering Research Center for Aquatic Animal Health Assessment, Shenzhen 518116, China

**Keywords:** CRP, Nile tilapia, Streptococcus agalactiae, Aeromonas hydrophila

## Abstract

**Simple Summary:**

C-reactive protein (CRP) is an acute-phase protein that can be used as an early diagnostic marker for inflammation. Few CRPs have been isolated from teleost, and the specific immunological functions and mechanism of fish CRP have not been well-studied. Therefore, in this research, a CRP gene from Nile tilapia was identified, and its roles during bacterial infection were investigated. The current results revealed that CRP participated in anti-bacterial immune response through agglutinating bacterial, regulating phagocytosis and inflammation. Hopefully, our data might be beneficial in further study to understand the protective mechanism of fish CRP against bacterial infection.

**Abstract:**

C-reactive protein (CRP) is an acute-phase protein that can be used as an early diagnostic marker for inflammation, which is also an evolutionarily conserved protein and has been identified from arthropods to mammals. However, the roles of CRP during the immune response of Nile tilapia (*Oreochromis niloticus*) remain unclear. In this study, a CRP gene from Nile tilapia (*On-CRP*) was identified, and its roles in response to bacterial infection were investigated in vivo or in vitro. *On-CR**P* was found to contain an open reading frame of 675 bp, encoding a polypeptide of 224 amino acids with the conservative pentraxin domain. On-CRP shares more than 50% of its identity with other fish species, and 30% of its identity with mammals. The transcriptional level of *On-CRP* was most abundant in the liver and its transcripts can be remarkably induced following *Streptococcus agalactiae* and *Aeromonas hydrophila* infection. Furthermore, in vitro analysis indicated that the recombinant protein of On-CRP improved phagocytic activity of monocytes/macrophages, and possessed a bacterial agglutination activity in a calcium-dependent manner. Both in vivo and in vitro experiments indicated that On-CRP could promote inflammation and activate the complement pathway. However, a direct relationship between CRP and several immune pathways could not be confirmed. The present data lays a theoretical foundation to further explore the mechanism of how CRP protects fish against bacterial infection.

## 1. Introduction

The C-reactive protein (CRP), first isolated from the pneumococcus-infected patients in 1930, was described initially as a third serologic fraction, or “fraction C” [1]. Subsequently, CRP was defined as a member of the “acute-phase reactant” group, because of its rapid increase in the serum of patients who exhibited inflammatory stimuli [2,3,4,5]. Further studies revealed that CRP was a hepatically derived, nonglycosylated pentraxin (PTX) possessing five identical subunits with calcium binding sites, and named based on its interaction with capsular polysaccharide (C-polysaccharide) of pneumococcus [6,7,8].

Functionally, CRP was also the first identified pattern-recognition receptor (PRR) [9,10] that has multiple immunological functions, including activating the complement system, promoting macrophages phagocytosis, interacting with FcγRI as well as a possible role in antigen presentation [6,11], which was considered as an inflammation marker and a major risk marker for cardiovascular disease as well [12]. CRP could be induced during inflammatory conditions including rheumatoid arthritis, infection, and some cardiovascular diseases [5,6] and CRP concentrations of serum would increase over 1000-fold during infection or suffering tissue damage [5].

Structurally, CRP is an evolutionarily conserved protein which has been identified from arthropods to humans, distinctively characterized by the typical PTX domains (HxCxS/TWxS) [8,13,14,15,16,17]. CRP is constitutive expressed in arthropods and molluscs, while working as an acute-phase protein (APP) in humans and most fish species [13]. Researchers suggest that CRP might have evolved as a component of the immune system’s development [13]. To date, few CRPs have been isolated from teleost and their bacterial agglutination and calcium-dependent lectin activity were recorded [14,17,18,19]. However, the specific immunological functions and mechanism of CRP in fish immunity have not been well-studied.

In this research, the CRP homolog was identified and characterized in Nile tilapia (*Oreochromis niloticus*), an important cultured freshwater fish in South China [20]. Moreover, the expression profiles and immunological roles of On-CRP during two common pathogenic bacteria, *Streptococcus agalatiae* [21] (Gram-positive bacteria), and *Aeromonas hydrophila* [22] (Gram-negative bacteria) infection were investigated. These data will expand our understanding of the roles of fish CRP during immune response against bacterial infection.

## 2. Materials and Methods

### 2.1. Ethical Statement

This study was performed in line with the principles of the Declaration of Helsinki. Approval was granted by the Ethics Committee of the Guangdong Ocean University (Date: 10 May 2019).

### 2.2. Fish Rearing

Healthy Nile tilapia (50 ± 10 g) were obtained from a fish farm in Zhanjiang, Guangdong, China. The fish were acclimatized in a 1000 L tank with 50 fish per cubic meter of water for >30 days. Fish were fed with commercial feed daily (3% of their body weight). The water temperature was maintained at 28 °C ± 0.5 °C. pH and dissolved oxygen were maintained within 7.3–7.8 and 5.0–6.0 mg/L, respectively. All fish for subsequent experiments were randomly selected.

### 2.3. RNA Extraction and cDNA Synthesis

Three healthy fish were anaesthetized with 3-aminobenzoic acid ethyl ester methanesulfonate (MS-222, Sigma, Darmstadt, Germany). Next, eight tissues including brain, gills, head kidney, intestine, liver, muscle, skin, and spleen, were collected and the total RNA was immediately extracted. RNAiso Plus (TaKaRa, Dalian, China) was used to extract the total RNA following the manufacturer’s protocol. The total RNA quantity and purity were verified via electrophoresis using 1.2% agarose gels and measured using a NanoDrop 2000 (Thermo Fisher Scientific, Waltham, MA, USA).

The RNA was reverse-transcribed via a PrimeScript^TM^ RT reagent kit with gDNA Eraser (TaKaRa, Dalian, China) following the manufacturer’s instructions. The cDNA was then diluted with redistilled water at a ratio of 1:50 for subsequent experiments.

### 2.4. Pathogenic Bacteria and Challenge

*Streptococcus agalactiae* (ZQ0910) was isolated from Nile tilapia and kept in the laboratory [23]. The preserved strain was first cultured in a fresh brain–heart infusion liquid medium at 28 °C overnight at 80 rpm, then collected via centrifugation (4000× *g*, 5 min), washed thrice in phosphate-buffered saline (PBS) and resuspended in PBS.

*Aeromonas hydrophila* was isolated from diseased fish and kept in the laboratory (unpublished data). The preserved strain was first cultured in Luria-Bertani (LB) liquid medium at 28 °C overnight at 80 rpm and prepared as mentioned above.

Subsequently, 100 fish were randomly collected and divided into two groups, namely, the *S. agalactiae* group and *A. hydrophila* group, with 50 fish per group. Then, all fish in each group were intraperitoneally injected with *S. agalactiae* (5 × 10^7^ CFU/mL, 100 µL per fish) [24] or *A. hydrophila* (5 × 10^7^ CFU/mL, 100 µL per fish), respectively. For each group, three fish were sacrificed at five different time points (0 h, 6 h, 12 h, 24 h, and 48 h) post challenge and five tissues including the brain, head kidney, intestine, liver, and spleen, were collected and RNA extracted and cDNA synthesized immediately from them.

### 2.5. Cloning and Sequence Analysis of On-CRP

All the primers (Appendix A) used in this study were designed with the NCBI Primer designing tool (https://www.ncbi.nlm.nih.gov/tools/primer-blast/ (accessed on 4 April 2021)). The predicted gene sequence of *On-CRP* was obtained from NCBI (https://www.ncbi.nlm.nih.gov/nuccore/XM_019364598.2/ (accessed on 4 April 2021)) and the completed open reading frame (ORF) sequence was amplified via polymerase chain reaction (PCR) with cDNA of liver and specific primers.

The multiple sequence alignments of putative CRP protein among various species were conducted using DNAMAN software (version 7.0, Lynnon Biosoft, San Ramon, CA, USA). The neighbor-joining (NJ) phylogenetic tree was constructed with MEGA software (version 6.0, Sudhir Kumar, PA, USA) with 1000 bootstrap replications. The potential signal peptide and transmembrane domain were predicted with SignalP (http://www.cbs.dtu.dk/services/SignalP/ (accessed on 10 April 2022)) and TMHMM (https://services.healthtech.dtu.dk/service.php?TMHMM-2.0 (accessed on 10 April 2022)), respectively. The molecular weight was predicted via the ProtParam tool (https://web.expasy.org/protparam/ (accessed on 10 April 2022)).

### 2.6. Quantitative Real-Time PCR (qRT-PCR)

The tissue distribution of *On-CRP* transcripts in healthy Nile tilapia and its expression patterns against bacterial infection was performed with TB Green^®^ Premix Ex *Taq*™ II (Tli RnaseH Plus) (TaKaRa, Dalian, China) and QuantStudio 6 and 7 Flex Real-Time PCR Systems (Thermo Fisher Scientific, Waltham, MA, USA) following the manufacturers’ instructions. *β**-actin*, *glyceraldehyde-3-phosphate dehydrogenase*(*gapdh*) and *elongation factor-1a*(*ef1a)* (Appendix A) were used as reference genes. Relative levels of *On-CRP* were calculated following Vandesompele’s [25] and Hellemans’s method [26]. All reactions were performed in triplicates.

### 2.7. Preparation of Recombinant On-CRP Protein (rGST-On-CRP)

Specific primers (Appendix A) with the restriction sites *EcoR* I and *Xho* I were used to amplify *On-CRP* ORF without the signal peptide domain. The DNA fragment was purified, digested and ligated with the digested pGEX-4T-1 plasmid (BT Lab, Wuhan, China) and then the ligated product was transformed into *Escherichia coli* BL21 (DE3) (TransGen, Beijing, China). The positive clone was verified via PCR and DNA sequencing and then cultured in fresh LB liquid medium containing ampicillin (100 µg/mL) at 37 °C until absorbance at OD_600_ reached 0.4–0.6. Subsequently, isopropyl-β-d-thiogalactoside (IPTG) was added to a final concentration of 0.5 mmol/L and continued to be cultured at 28 °C for 8 h.

The protein was then purified by a GST-tag protein purification kit (Beyotime, Shanghai, China) and then analyzed by 12% SDS-PAGE and Coomassie Blue staining and Western blot as mentioned below. Additionally, an extra completed pGEX-4T-1 plasmid (BT Lab, Wuhan, China) was transformed to *E. coli* BL21 (DE3) and then the GST-tag protein was prepared as mentioned above for subsequent experiments.

### 2.8. Western Blot

The purified rGST-On-CRP was loaded on 12% SDS-PAGE and transferred to a PVDF membrane (IPVH00010, Merck, Darmstadt, Germany) and then blocked with QuickBlock™ Blocking Buffer for Western Blot (Beyotime, Shanghai, China) at 25 °C for 15 min. The membrane was then incubated with anti-GST-tag antibodies (AG768, Beyotime, Shanghai, China), diluted at a ratio of 1:1000 in QuickBlock™ Primary Antibody Dilution Buffer for Western Blot (Beyotime, Shanghai, China) and incubated at 25 °C for 1 h.

Then the membrane was washed three times in TBST (TBS + 0.1% Tween 20) and incubated with secondary antibody HRP-labeled goat anti-rabbit IgG (H + L) (A0208, Beyotime, Shanghai, China), diluted at a ratio of 1:2000 in QuickBlock™ Secondary Antibody Dilution Buffer for Western Blot (Beyotime, Shanghai, China) at 25 °C for 30 min. Finally, the membrane was washed three times in TBST, and the antigen–antibody complexes were detected via the enhanced chemiluminescence method (P0018S, Beyotime, Shanghai, China).

### 2.9. Nile tilapia Head Kidney Leukocytes (HKLs) and Monocytes/Macrophages (MO/MΦ) Preparation

Nile tilapia HKLs were prepared as previously described [24]. Briefly, three healthy fish were sacrificed as mentioned above and then the head kidney of each fish was separated, cut, and sieved through a 40 μm stainless nylon mesh (Greiner Bio-OneGmbH, Frickenhausen, Germany) for collecting the cells which were suspended in Leibovitz’s L-15 Medium (Thermo Fisher Scientific, Waltham, USA). The cells were very gently added to 34%/51% percoll (Solarbio, Beijing, China) and centrifuged at 400× *g* for 40 min. Then, the leukocytes between the 34% percoll layer and the 51% percoll layer were carefully aspirated. The leukocytes were gently washed with PBS, centrifuged at 500× *g* for 10 min for collection, and then resuspended and cultured in Leibovitz’s L-15 Medium (Thermo Fisher Scientific, Waltham, MA, USA). A cell counting plate was utilized to count the cells from each fish.

MO/MΦ were then isolated following the method described by Mu [27]. Concisely, the HKLs were suspended with Leibovitz’s L-15 Medium to a density of 5 × 10^5^ cells/mL and cultured in 12-well plates at 25 °C for 24 h. Subsequently, the non-adherent HKLs were removed through the exchange of the cell medium twice and the adherent cells (MO/MΦ) were retained. Next, MO/MΦ was separated by trypsin digestion, collected by centrifugation (1500× *g*, 2 min), washed twice and resuspended in PBS for subsequent experiments. A cell counting plate was used to count the cells.

### 2.10. Phagocytosis Assay

The bacteria (*S. agalactiae* and *A. hydrophila*) were prepared as mentioned above and suspended in Na_2_CO_3_ (0.1 M, pH 9.5) at 10^8^ CFU/mL, then the fluorescein isothiocyanate (FITC, Solarbio, Beijing, China) was added into bacteria to the final concentration at 0.5 mg/mL and then incubated for 0.5 h in dark at 25 °C to label thorough [28]. Subsequently, the bacteria were washed three times in PBS to remove the unlabeled FITC, suspended in PBS and then the MO/MΦ were added to the final volume at 480 µL and the final individual ratio of bacteria (10^6^ CFU/mL) and MO/MΦ (10^6^ cell/mL) reached 1:1. Meanwhile, 20 µL of rGST-On-CRP (49.5 μg (1 nmol), dissolved in PBS), GST-tag (26 μg (1 nmol), dissolved in PBS) or PBS were added to each solution and then incubated for 0.5 h in the dark at 25 °C [28,29]. The non-ingested bacteria were removed by centrifugation and the phagocytosis was assayed via a flow cytometer. Each test was performed with three replicates as well as every fluorescence data is limited to a gate to ensure accuracy.

### 2.11. Bacterial Agglutination Capacity Assay

The bacteria (*S. agalactiae* and *A. hydrophila*) were prepared as mentioned above and incubated with rGST-On-CRP, GST-tag, CaCl_2_ or EDTA with the final concentration of bacteria at 10^7^ CFU/mL, rGST-On-CRP and GST-tag at 2 μM, CaCl_2_ at 10 mM and EDTA at 20 mM. Then the solutions were incubated for 0.5 h at 25 °C, observed and photographed using a ZEISS Axioscope 5 microscope (Zeiss, Jena, Germany).

### 2.12. On-CRP Function and Molecular Mechanism Assay In Vivo

A total of 350 healthy fish were randomly divided into seven groups (50 fish per group), namely, PBS group, *S. agalactiae* + PBS group, *S. agalactiae* + GST group, *S. agalactiae* + CRP group, *A. hydrophila* + PBS group, *A. hydrophila* + GST group, and *A. hydrophila* + CRP group. For the PBS group, 100 μL PBS was intraperitoneally injected into each fish as a control. For the *S. agalactiae* + PBS group, 100 μL *S. agalactiae* (5 × 10^7^ CFU/mL, dissolved in PBS) was injected while the *S. agalactiae* + GST and *S. agalactiae* + CRP group were co-injected with 4 nmol of GST-tag protein (104 μg) and rGST-On-CRP (197.2 μg) dissolved in the same 100 µL of PBS with *S. agalactiae*, respectively. The parallel processing was also performed among three *A. hydrophila* challenge groups with 100 μL *A. hydrophila* (5 × 10^7^ CFU/mL).

From each of the six challenged groups, the head kidney, liver and spleen tissues from three fish were collected for total RNA extraction as mentioned above at four time points (6 h, 12 h, 24 h, and 48 h) after injection. Moreover, three fish of PBS group at 0 h were collected as a control (Mock) as well.

In addition, from the six challenged groups, 10 mg tissues (liver and spleen) were collected at 48 h as mentioned above, broken and homogenized in 1 mL of PBS, then diluted with PBS at a ratio of 1:500. Subsequently, 50 μL solutions of the tissue homogenates were continued cultured on a BHI plate (*S. agalactiae* challenge groups) or LB plate (*A. hydrophila* challenge groups) at 28 °C for 24 h to assess the bacterial burden, as well as the *16s rRNA* genes of 10 random picked clones from each plate were sequenced to verify the bacterial species. Each group was repeated in six parallels.

Daily statistical mortality, as well as the survival rate (SR) from the challenge test for seven days, were calculated following our previous iterative calculation formula [24].
$$\mathrm{SR}=\left(1-\frac{Fatality\;fish}{\;Survival\;fish_{\left(previous\;day\right)}-sampled\;fish}\right)\times {\mathrm{SR}}_{\left(previous\;day\right)}\times 100\%$$

The roles of On-CRP in regulating immune response to bacterial infection were further clarified by assessing numerous immune-related genes (Appendix A), including inflammatory factors, complement, interferon, and four key genes of four immune pathways.

### 2.13. On-CRP Function on Nile tilapia HKLs Response to Stimulation Assay

To further evaluate the role of On-CRP on regulating the HKLs response to stimulation or inflammation, lipopolysaccharide (LPS, dissolved in Leibovitz’s L-15 Medium, 1 mg/mL) (Beyotime, Shanghai, China) and lipoteichoic acid (LTA, dissolved in Leibovitz’s L-15 Medium, 1 mg/mL) (Shanghai Yuan Ye Bio, Shanghai, China) were employed as stimulants and then the two inflammation models were established. Specifically, LPS simulates Gram-negative bacteria and LTA simulates Gram-positive bacteria.

Three healthy Nile tilapia were collected and the HKLs of each fish were prepared as mentioned above (working as three biological replicates). HKLs were cultured in 12-well plates at 10^5^ cells per well and LPS/LTA was added to the HKL well to make a final concentration of 5 μg/mL [30]. Additionally, the same volume of Leibovitz’s L-15 Medium was added to the control group to simulate a resting state.

rGST-On-CRP, GST-tag protein or the same volume of PBS (working as a control) was added to each group at a final concentration of 2 μM, respectively. Then, the HKLs were collected via centrifugation (1500× *g*, 2 min) at 10 min, 30 min, 3 h, and 6 h post addition. Additionally, three samples of control groups at 0 min were collected as a control (Mock) as well.

RNA was extracted and reverse-transcribed as explained before. qRT-PCR was performed to assess the potential functions and molecular mechanisms of On-CRP in regulating immune responses using various related genes. Appendix A lists all primers.

### 2.14. Drawings and Statistical Analysis

Adobe Photoshop CC (San Jose, CA, USA) and Adobe Illustrator (San Jose, CA, USA) were used for drawings and final panel designing. All data were presented as means ± standard deviations (SD). The one-way ANOVA tests (Tukey HSD test) and Student’s *t*-tests were used to analyze the significant difference via Prism software (version 8.0, GraphPad Software, Inc., San Diego, CA, USA). Different letters and asterisks illustrate statistically significant differences (*p* < 0.05).

## 3. Results

### 3.1. Characteristics of On-CRP

The ORF of *On-CRP* is 675 bp that encodes a putative protein of 224 amino acids, including a 15 amino acid residues signal peptide domain of the N-terminal while without transmembrane domain. The predicted molecular mass of On-CRP is 23.56 kDa, and its theoretical pI is 8.29. Multiple sequence alignments indicated that the deduced amino acid sequence of On-CRP contains a PTX domain and a series of conservation of functionally critical amino acid residues. Additionally, the BLAST analysis indicated that the deduced amino acid sequence is homologous with other CRP, which is about 50% identical to other fish species and more than 30% identical to mammals (Figure 1A). Phylogenetic analysis indicated that On-CRP initially clustered with other fish lineages, and then clustered with amphibians and mammals CRP (Figure 1B).

### 3.2. Expression Characteristics of On-CRP among Different Tissues

qRT-PCR was employed to assess the relative expression level of *On-CRP* among different tissues of healthy tilapia. The highest *On-CRP* expression was observed in the liver, whereas it was relatively high in the spleen (Figure 1C).

### 3.3. Expression Characteristics of On-CRP after Bacteria Challenge

The transcriptional levels of *On-CRP* were significantly promoted (*p* < 0.05) in all examined tissues (brain, head kidney, intestine, liver, and spleen) post bacterial infection. Additionally, the expression peak of *On-CRP* was reached at 12 h in the head kidney and liver while peaked until 24 h in the brain, intestine and spleen after the *S. agalactiae* challenge (Figure 1D(a)). However, the expression peak was rapidly reached at 6 h in the liver while at 12 h in the spleen and until 24 h in the rest organs post *A. hydrophila* challenge (Figure 1D(b)).

### 3.4. Effects of rOn-CRP against Bacteria In Vitro

The effects of On-CRP against bacteria were determined by assessing phagocytosis and bacterial agglutination capacity to *S. agalactiae* and *A. hydrophila*. The recombinant protein with a predicted molecular weight of 49.3 kDa (rGST-On-CRP) was prepared and then confirmed via SDS-PAGE and Western blot (Figure 2).

After treatment with rGST-On-CRP (2 μM), the number of bacteria swallowed by MO/MΦ were significantly improved than that in blank group (PBS treated) or the negative control (2 μM GST-tag protein treated) (Figure 3A).

As shown in Figure 3B, no evident bacterial agglutination activity was observed post GST-tag or rGST-On-CRP treatment (first and fourth column). However, some mild agglutinated bacteria were detected post CaCl_2_ treatment (second column). Moreover, the combination of CaCl_2_ and rGST-On-CRP agglutinated bacteria more distinctly (second and fifth column), while the combination of CaCl_2_ and GST-tag didn’t promote the agglutination degree (second and third column). And finally, the agglutination was recovered by EDTA (sixth column).

### 3.5. Effects of rOn-CRP against Bacterial Infection In Vivo

The SR of the PBS group, *S. agalactiae* + PBS group, *S. agalactiae* + GST group, *S. agalactiae* + CRP group, *A. hydrophila* + PBS group, *A. hydrophila* + GST group, and *A. hydrophila* + CRP group were 100%, 49%, 46%, 56%, 37%, 40%, and 43%, respectively, without distinct difference among challenge groups (Figure 4A). Moreover, only the bacterial burden of the spleen during the *S. agalactiae* challenge was significantly reduced (Figure 4B).

The functions of rOn-CRP against bacterial infection were further determined by detecting numerous immune-related genes via qRT-PCR. The expression levels of inflammatory factors, complement, interferon and key genes of four immune-related pathways all increased after bacterial infection (Figure 5, Figure 6 and Figure 7).

Compared with that of the PBS and GST groups, the expression levels of inflammatory factors of CRP groups were mainly significantly increased (*p* < 0.05) at series of time points during both *S. agalactiae* and *A. hydrophila* infection (Figure 5). The expression levels of complement (*C3*, *C5*) and interferon (*IFN-γ1*) of CRP groups were significantly increased in the liver and spleen, but decreased in the head kidney during *S. agalactiae* infection (Figure 6). Additionally, the expression levels of complement and interferon were improved by rOn-CRP as a whole during *A. hydrophila* infection (Figure 6). However, four key genes, *P65*, *P38*, *MyD88* and *STAT3*, that belong to different immune pathways, were minimally induced by rGST-On-CRP, except at few time points of them against *A. hydrophila* infection (Figure 7).

### 3.6. Effects of rOn-CRP on Nile tilapia HKLs Response to Stimulation

To further confirm the roles of rOn-CRP on immune response, the expression patterns of inflammatory factors and key genes of immune pathways in HKLs were detected and as shown in Figure 8, the inflammatory factors were promoted by rOn-CRP in both resting state (control group) and two inflammatory models (Figure 8A). However, the expression levels of *P65*, *P38*, *MyD88* and *STAT3* were not remarkably induced in inflammatory models but improved or reduced in resting state (Figure 8B).

## 4. Discussion

CRP protects the host against a variety of acute and chronic inflammation, and plays roles in both innate and adaptive immune responses [5,6,11,13]. In this study, we identified and characterized a CRP homolog (*On-CRP*) from Nile tilapia.

On-CRP is a secreted protein that contains a conserved PTX domain, which shares over 50% of its identity with other fish CRPs [13,17,29,31], and more than 37% of its identity with mammalian CRPs [13,15,32], implying the conservation and importance of this molecule during evolution. Similar to mammals and some fish species CRP [11,17,29,31,33], the highest transcriptional level of *On-CRP* was detected in the liver, whereas lower levels were observed in other tissues, suggesting that *On-CRP* was mainly synthesized in the liver. In contrast, the highest transcriptional level of *CRP* has been observed in the spleen of several species of fish [34,35], suggesting that CRP might have experienced several distinct evolutional processes in fish lineages.

After *S. agalactiae* and *A. hydrophila* challenged, qRT-PCR results showed that *On-CRP* expressions were up-regulated in all tested tissues, such as the major synthetic organ liver and the major immune-related tissues such as the head kidney and spleen [36], implying that *On-CRP* might participated in the immune response against bacterial infection. Our findings suggest that the expression of *On-CRP* might reach peaks earlier at 6–12 h following *A. hydrophilia* infection compared to *S. agalactiae* infection where peaks of the expression of *On-CRP* are observed from 12 h to 24 h. This might imply the expression of CRP is noticed earlier in Gram-negative bacteria than in Gram-positive bacteria as was found in earlier studies [34].

To further explore the roles in response to bacterial infection, the rGST-On-CRP was prepared. Consistent with other fish CRPs [29], the promoting effect of rGST-On-CRP on MO/MΦ phagocytosis activity was observed. However, the complement-mediated opsonophagocytosis of MO/MΦ was suppressed in ayu [17] and the studies focusing on fish MO/MΦ phagocytosis are rare [29]. Therefore, to further understand the functions of fish CRP functions on MO/MΦ, there is a need for further studies. In addition, Ca^2+^ is necessary for rGST-On-CRP agglutinating bacteria, especially *A. hydrophila*, similar to other fish CRPs [17,34]. However, it has been suggested that Ca^2+^ at 10 mM is the concentration that can slightly agglutinate these two bacteria. In this study (data not shown), a lower (5 mM) or a higher concentration (50 mM) Ca^2+^ has been tested and found to have the potential to agglutinate these bacteria. However, there is a need for further study to elucidate the optimum Ca^2+^ concentration necessary for Tilapia CRP [17,34].

In mammals, CRP could increase dramatically during injury, infection, and inflammation [5]. In this study, the results of in vivo experiments showed that rGST-On-CRP only provided limited protection against bacterial infection (SR and bacterial burden), and the expression of inflammatory factors, complement, and interferon were induced at the transcriptional level.

In fact, however, the exact pro-inflammatory or anti-inflammatory roles of CRP are still controversial in mammals [37]. That notwithstanding, the present results suggest that rOn-CRP might mainly enhance inflammation at the early stages of *S. agalactiae* infection (6 h and 12 h post-infection) but inhibit inflammation in later hours (i.e., 24 h and 48 h) post-infection as evidenced by the increase in pro-inflammation factors *IL-1β* and *TNF-α* and anti-inflammation factors *IL-10* and *TGFβ,* respectively. Similar results have been observed in human macrophages [38] and mudskipper [29]. However, the regulation effects of rOn-CRP on inflammatory factors against *A. hydrophila* infection were more ambiguous that both pro-inflammatory and anti-inflammatory factors were promoted as a whole and we couldn’t explain it yet.

In mammals, the complement pathway activated by CRP has been extensively investigated [5,6,10]. Similar findings were also obtained in our study. The transcripts of *C3* and *C5*, the core members of the complement pathway [5], were significantly up-regulated in the liver and spleen, but the trend in the head kidney was opposite during *S. agalactiae* infection. The results can be explained by those complements were mainly synthesized in the liver, the increase in head and kidney doesn’t mean enough actual significance.

It has been well-known that human CRP promotes the activation of the NK-κB pathway [13,39] and the MAPK pathway [40], and interacts with the TLR [41] and STAT3 pathway [42]. However, the relationship between these pathways and CRP has not been established in fish. Our data indicated that the responses of immune pathways during infection might be minimally modulated by rOn-CRP, inconsistent with the research of human CRP. Thus, the in vitro HKLs stimulation experiments were performed and the results basically coincide with the in vivo data. These data emphasize the need to revisit the direct contact of fish CRP and these immune pathways, and this will be our interest in future study.

FcγR, a receptor for CRP in mammals, mediates multiple stimulus signals, which suggests a possible role in antigen presentation and contributes to the process of ageing [6,38,43,44], however, this is missing in fish [17,45]. Similarly, the transmembrane FcγR was also lost in tilapia genome [46]. These results imply that the roles of CRP in mediation adaptive immunity and ageing seem to be associated with the evolution from fish to mammals.

## 5. Conclusions

In summary, our study identified a tilapia C-reactive protein homolog (On-CRP) and explored its roles in an immune response against bacterial infection. On-CRP plays roles in bacterial agglutination, phagocytosis, and modulating inflammation during bacterial infection. The current data are beneficial to further studying the protective mechanism of fish CRP against pathogens.

## Figures and Tables

**Figure 1 biology-11-01149-f001:**
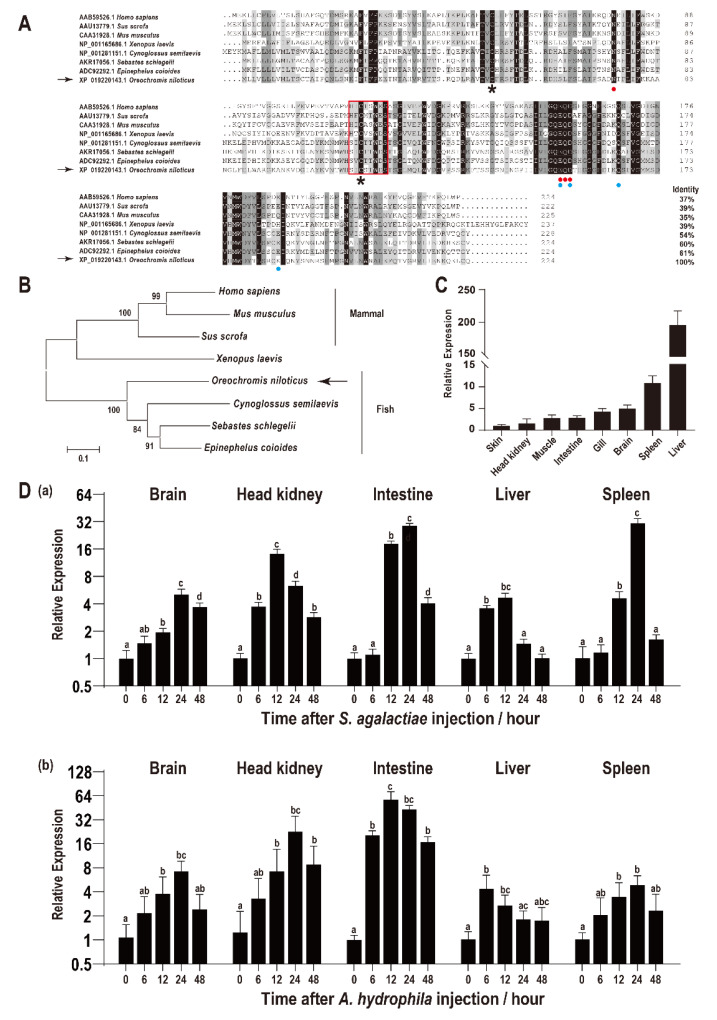
(**A**) Multiple sequence alignments of CRP among different species. The percentage value following each amino acid sequence indicates overall sequence identity between On-CRP and other sequences. Similar residues are marked in grey and consensus residues are marked in black. The red box indicates the PTX domain. The asterisks below the alignment indicate the two conserved cysteine residues. The two calcium ion binding sites were indicated by red points (site 1) and blue points (site 2), respectively. (**B**) Phylogenetic tree of On-CRP among different species constructed using the neighbour-joining method. The numbers at the nodes between the branches point to the bootstrap confidence values with 1000 replicates. (**C**) Relative expression levels of *On-CRP* mRNA among different tissues in healthy Nile tilapia were assessed via qRT-PCR. All values are the mean ± SD; *n* = 3. The expression level of *On-CRP* in the skin was set as 1. (**D**) Expression patterns of *On-CRP* mRNA in different tissues (brain, head kidney, intestine, liver and spleen) at a series of time points after *S. agalactiae* (**a**) and *A. hydrophila* (**b**) injection as detected via qRT-PCR. The expression level of *On-CRP* at 0 h was set as 1.00 to calculate the relative expression of the other time points. All values are the mean ± SD, *n* = 3. Different letters indicate significant difference (*p* < 0.05).

**Figure 2 biology-11-01149-f002:**
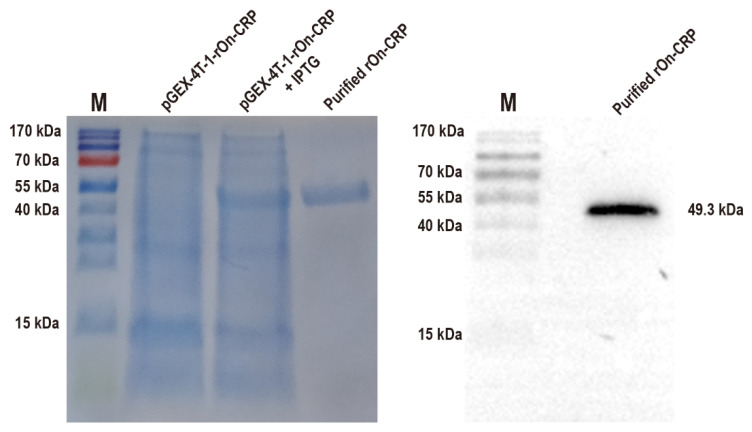
SDS-PAGE of the total proteins of the uninduced and induced bacterial strain contain prokaryotic recombinant vector pGEX-4T-1-rOn-CRP and the purified recombinant protein with the predicted molecular weight 49.3 kDa (**left**). Western blot of the purified rGST-On-CRP, detected via an anti-GST-tag antibody (**right**) The uncropped western blot figure was presented in Appendix A.

**Figure 3 biology-11-01149-f003:**
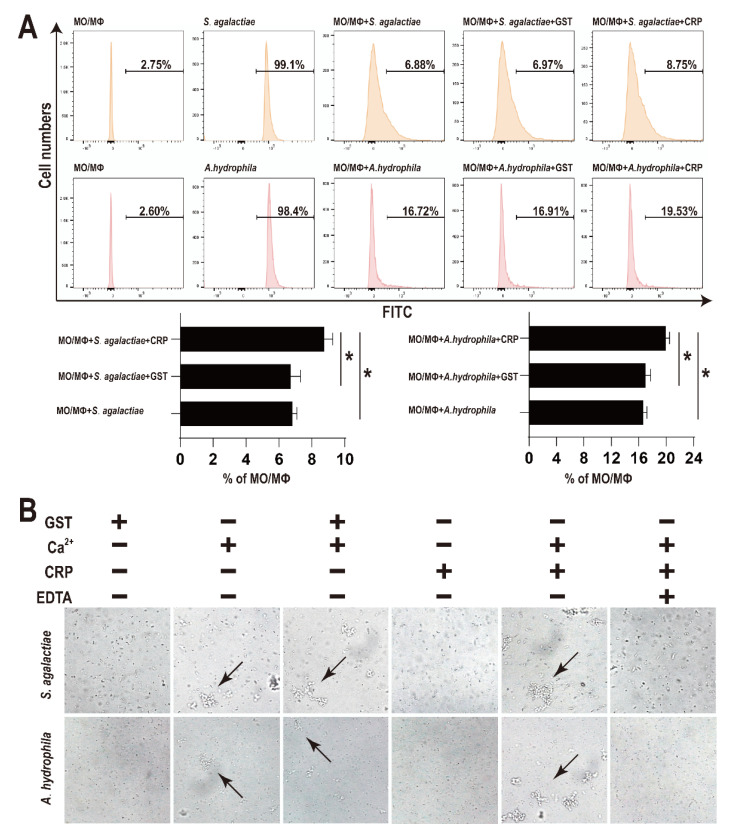
(**A**) Effects of rOn-CRP on phagocytosis activity of monocytes/macrophages. The FITC labelled bacteria were incubated with MO/MΦ and rGST-On-CRP (2 μM)/GST-tag protein (2 μM) for 0.5 h at room temperature. Significant differences are indicated by asterisks (*p* < 0.05). (**B**) Bacterial agglutination caused by rOn-CRP. The bacteria were incubated with GST-tag protein (2 μM), CaCl_2_ (10 mM), rGST-On-CRP (2 μM) and EDTA (20 μM) for 0.5 h at room temperature. Arrows point to the agglutinated bacteria. All photographs were observed at 400× magnification.

**Figure 4 biology-11-01149-f004:**
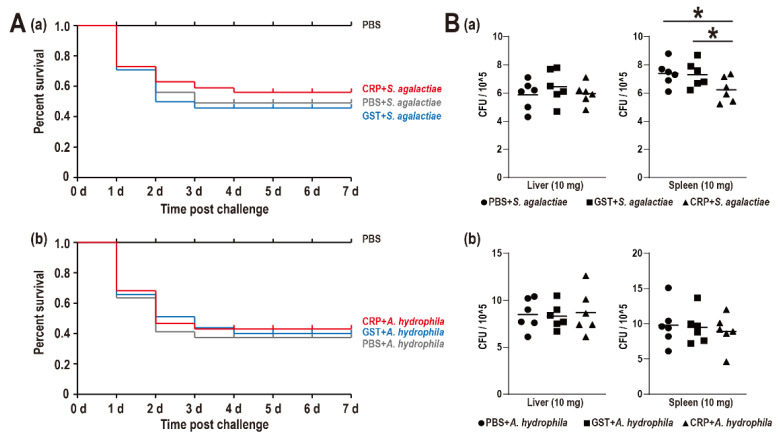
(**A**) Survival rates of Nile tilapia after *S. agalactiae* (**a**) and *A. hydrophila* (**b**) infection. Daily statistical mortality was calculated for 7 days, *n* = 50 for each group. 100 μL PBS for PBS group, *S. agalactiae* (5 × 10^7^ CFU/mL) dissolved in 100 μL PBS for *S. agalactiae* + PBS group, *S. agalactiae* (5 × 10^7^ CFU/mL) with GST-tag protein (4 nmol)/rGST-On-CRP (4 nmol) dissolved in the 100 μL PBS for *S. agalactiae* + GST group/*S. agalactiae* +CRP group. The parallel processing was also performed among three *A. hydrophila* challenge groups with 100 μL *A. hydrophila* (5 × 10^7^ CFU/mL). (**B**) Bacterial burden in liver and spleen of Nile tilapia post *S. agalactiae* (**a**) and *A. hydrophila* (**b**) infection at 48 h. All values are the mean ± SD; *n* = 6. Significant difference is indicated by asterisks (*p* < 0.05).

**Figure 5 biology-11-01149-f005:**
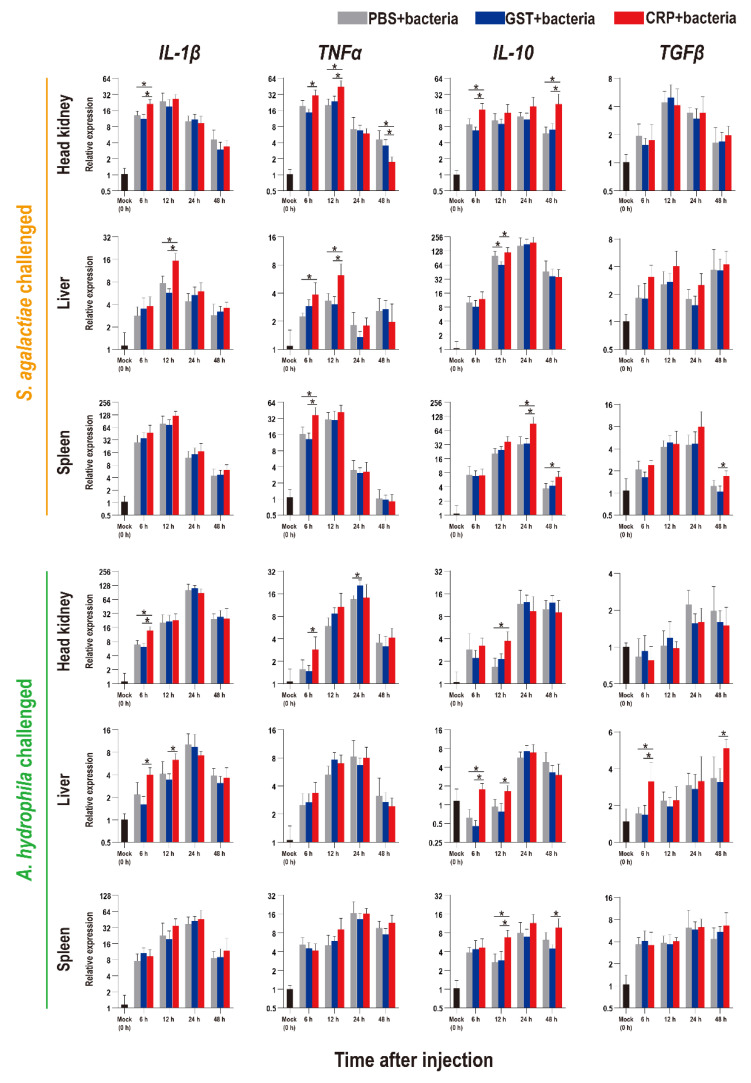
Expression patterns of inflammatory factors as assessed via qRT-PCR at different time points post *S. agalactiae* and *A. hydrophila* infection. All values are the mean ± SD; *n* = 3. Significant difference is indicated by asterisks (*p* < 0.05).

**Figure 6 biology-11-01149-f006:**
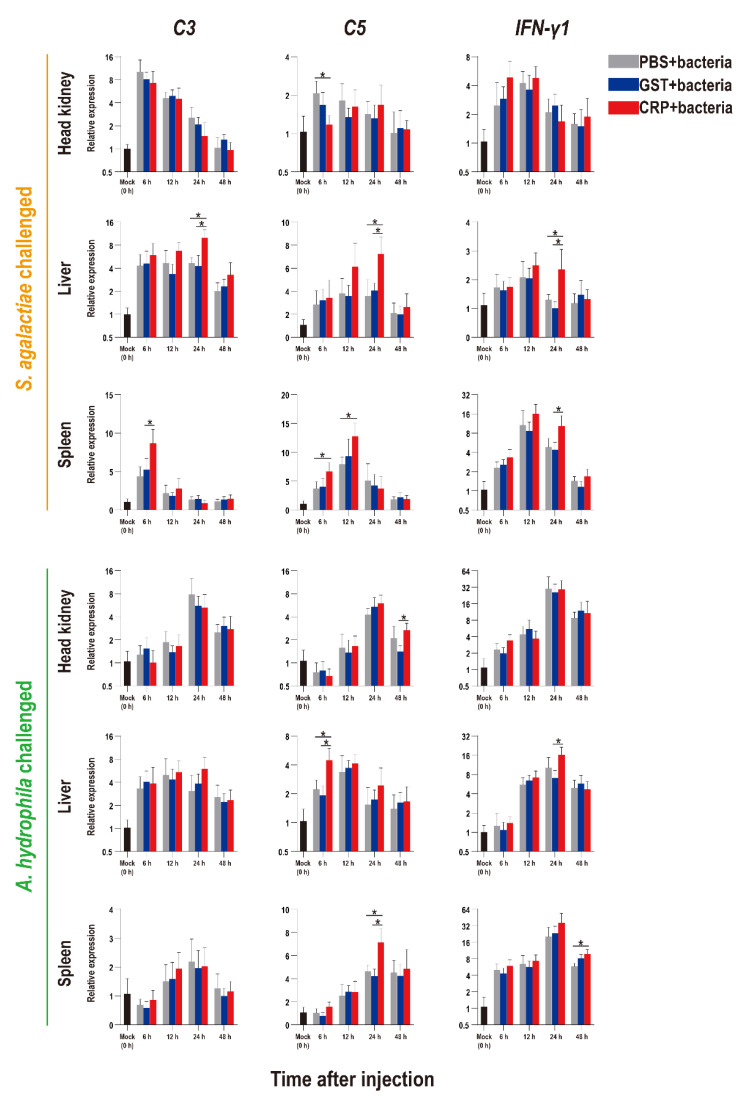
Expression patterns of complements (*C3* and *C5*) and interferon (*IFN-γ1*) as assessed via qRT-PCR at different time points post *S. agalactiae* and *A. hydrophila* infection. All values are the mean ± SD; *n* = 3. Significant difference is indicated by asterisks (*p* < 0.05).

**Figure 7 biology-11-01149-f007:**
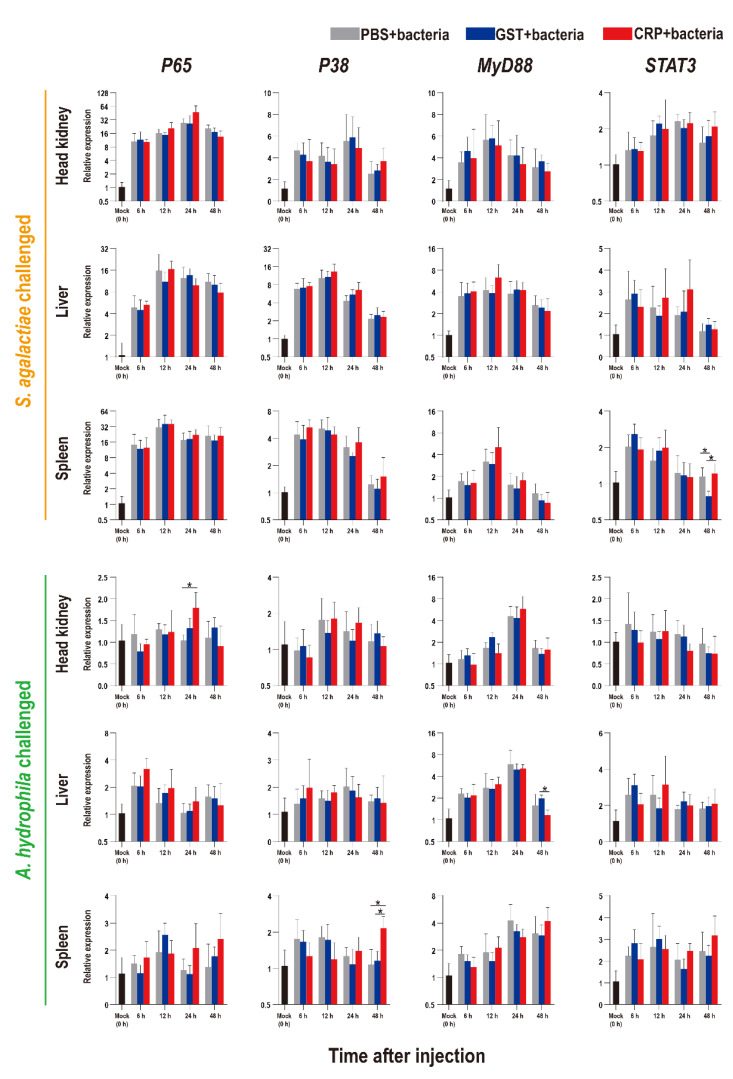
Expression patterns of key genes of four immune-related pathways as assessed via qRT-PCR at different time points post *S. agalactiae* and *A. hydrophila* infection. All values are the mean ± SD; *n* = 3. Significant difference is indicated by asterisks (*p* < 0.05).

**Figure 8 biology-11-01149-f008:**
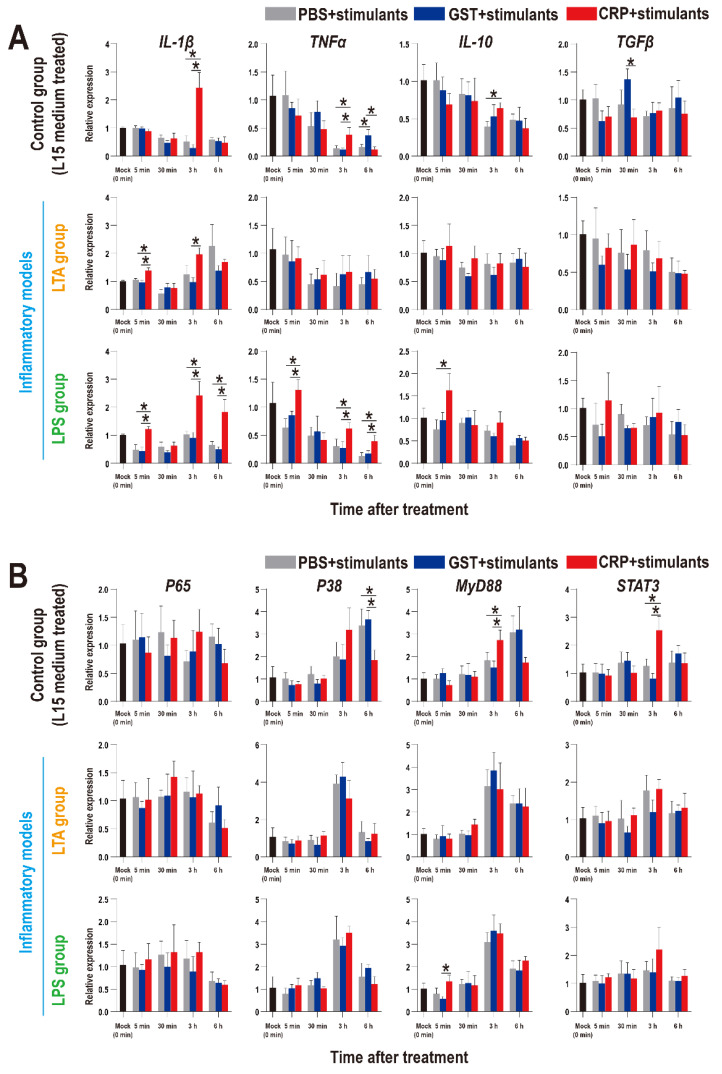
Expression patterns of inflammatory factors (**A**) and key genes of four immune-related pathways (**B**) in HKLs as assessed via qRT-PCR at different time points with different treatments. All values are the mean ± SD; *n* = 3. Significant difference is indicated by asterisks (*p* < 0.05).

## Data Availability

Data is contained within the article or Appendix A.

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
