# Peer review of "CRP Involved in Nile Tilapia (Oreochromis niloticus) against Bacterial Infection"

_biology, 2022, doi:10.3390/biology11081149_

Round 1

Reviewer 1 Report

I was honored to review the manuscript entitled “CRP involved in Nile tilapia (Oreochromis niloticus) against bacterial infection ” submitted to Biology Journal. C-reactive protein (CRP) is an annular (ring-shaped) pentameric protein found in blood plasma, whose circulating concentrations rise in response to inflammation. It is an acute-phase protein of hepatic origin that increases following interleukin-6 secretion by macrophages and T cells. This research article could help to open the wide insight about the role of CRP against bacterial pathogens in fish.I would like to thank authors for preparing this research article in the perfect way. The study presents high quality and well written.

In conclusion I believe that the present manuscript should be accepted in present form.  

Author Response

Thanks to reviewer for your highly comments.

Reviewer 2 Report

In this paper, C-reactive protein (CRP) of Nile tilapia (Oreochromis niloticus) was recombinantly expressed, and its role against Streptococcus agalactiae and Aeromonas hydrophila infection was explored. Overall, very interesting research results were described in the paper.

Questions ang suggestions,

1.      The challenge dose was 5 × 106 CFU, and more information should be shown in the article if a pre-experiment was done.

2.      The concentration of the stimuli is important and more information on the determination of the concentrations of LPS, LTA, GST, rOn-CRP should be presented in the article.

3.      Whether the labeling efficiencies of S. agalactiae and A. hydrophila are equivalent and the impact on phagocytosis activity could be explained in the Discussion.

4.      Some pictures still have room for adjustment. For example, negative controls could be added to the WB results in Figure 2, and bars should be included in Figure 3B.

Reviewer 3 Report

This manuscript has many English problems; some are listed below, but they are not all. I strongly suggest that the manuscript should be reviewed, check and corrected by English native speakers before re-submission.

1.        Line 22: mediation bacterial agglutination, regulation inflammation

2.        Line 26: is an “evolutionarily protein” has been identified from

3.        Line 30: change “peptide” to “polypeptide”

4.        Line 31: shares

5.        Line 32: remarkably

6.        Line 39: add “how” before CRP.

7.        Line 63: “component of evolution” is hard to understand.

8.        Line74: understanding of

9.        Line 195-196: change “assumed” to “assayed”.

10.     Line 208: Should “peritoneal” change to “intraperitoneally”?

11.     Line 385: I don’t think “generation” is properly used in this sentence.

12.     Line 389-394: please rewrite this long sentence to make it more readable.  

13.     Line 415: “amphibious”?

14.     Line 220: change “broken and dissolved” to “homogenized”.

It seems that many listed references don’t match the main text. Some errors are listed below, but I think they are not all. Please thoroughly check the reference list before re-submission.

15.     Line 135: Vandesompele’s [28] and Hellemans’s method [29].

16.     Line 227: the reference [33] doesn’t match the main text.

17.     Line 179: Mu [32].

18.     Line 414: mudskipper [34]

Other comments/problems:

19.     Lines 109-110: Please explain why 5×107 CFU/mL was used for injection.

20.     Line 222: Do the “BHI tablet” and “LB tablet” mean “BHI plate” and “LB plate”?  

21.     The meaning of the number in figure 1 B should be explained in figure legend.

22.     The legend of figure 2 could be described in more detail, for example, antibody used for the right figure could be indicated.

23.     The recombinant protein On-CRP is actually a GST fusion protein; therefore, it would be clearer if it is indicated as rGST-On-CRP.

24.     Figure 3B shows that Ca2+ alone could induce bacteria agglutination, and therefore the results of Figure 3 could not support the conclusion in lines 396-398 (as well as in line 35) that rOn-CRP promoted bacteria agglutination and Ca2+ is necessary for rOn-CRP agglutinating bacteria.      

25.     The legends of figures 6-8 should be described in more detail, as these immune genes were detected in tilapia injected with bacteria + PBS, bacteria + GST, or bacteria + CRP, but not injected with bacteria only.

26.     Line 335-336: What are “typical factors of immune pathways”?

27.     Line 405-406: I don’t understand “at microscopic level”

28.     Line 226: What is “daily statistical morbidity”?

29.     Line 228: Why the formula of survival rate is so complicated? According to the corresponding author’s previous paper (Front Immunol. 2020; 11: 1140.), the formula was simpler and calculated as SR = (surviving fish÷total fish no) × 100%.

Round 2

Reviewer 3 Report

I am not a native English speaker, but I can tell that this revised manuscript still has English problems. Some problems are listed below, but they are not all. I strongly suggest that the authors should send the manuscript to an English editing service.   

1.      Line 27: “CRP is an evolutionarily conserved protein has been identified”

2.      Line 150: “via induction the strain”

3.      Line 180: “isolated performed”

4.      Lines 182-183: “via Leibovitz’s L-15 Medium suspended twice as well as the adherent MO/MΦ were retained”

5.      Lines 223 and 225: please change “tablet” to “plate”. As “tablet” is used to prepare “plate”, and the bacteria is spread onto the “plate” not “tablet”.

6.      Line 277: “The transcriptional levels of On-CRP significantly increased”.

7.      Lines 309-311: hard to read.

8.      Line 401: please change “suggested” and “been” to “changing” and “be”

9.      Lines 445-446: “was not been…either” “implying”   
